



# Laboratory evaluation of particle size-selectivity of optical low-cost particulate matter sensors

Joel Kuula[1], Timo Mäkelä[2], Minna Aurela[1], Kimmo Teinilä[1], Samu Varjonen[3], Óscar González[4], and Hilkka Timonen[1]

[1]Atmospheric Composition Research, Finnish Meteorological Institute, Helsinki, FI-00560, Finland
[2]Climate System Research, Finnish Meteorological Institute, Helsinki, FI-00560, Finland
[3]Department of Computer Science, University of Helsinki, Helsinki, FI-00560, Finland
[4]Fab Lab Barcelona, Institute of Advanced Architecture of Catalonia, Barcelona, 08005, Spain

*Correspondence to*: Joel Kuula (joel.kuula@fmi.fi)

**Abstract.** Low-cost particulate matter sensors (PM) have been under investigation due to their prospective nature regarding spatial extension of measurement coverage. While majority of the existing literature highlights that low-cost sensors can be useful in achieving this goal, it is often reminded that the risk of sensor misuse is still high, and that the data obtained from the sensors is only representative of the specific site and its ambient conditions. This implies that there are underlying reasons yet to be characterized which are causing inaccuracies in sensor measurements. The objective of this study was to investigate the particle size-selectivity of low-cost sensors. Evaluated sensors were Plantower PMS5003, Nova SDS011, Sensirion SPS30, Sharp GP2Y1010AU0F, Shinyei PPD42NS, and Omron B5W-ld0101. The investigation of size-selectivity was carried out in laboratory using a novel reference aerosol generation system capable of steadily producing monodisperse particles of different sizes on-line. The results of the study showed that none of the low-cost sensors adhered exactly to the detection ranges declared by the manufacturers, and moreover, cursory comparison to a mid-cost aerosol spectrometer (GRIMM 1.108) indicated that the sensors could only achieve independent responses for 1-2 size bins whereas the spectrometer could sufficiently characterize particles with 15 different size bins. These observations provide insight and evidence to the notion that particle size-selectivity may have an essential role in the error source analysis of sensors.

## 1 Introduction

Recent emergence of low-cost sensors has enabled new possibilities in traditional air quality monitoring (Kumar et al., 2015; Morawska et al., 2018; Snyder et al., 2013). As a result of low unit cost and compact size, sensors can be deployed to the field in much higher quantities than before and thus enabling higher resolution spatiotemporal data. Few studies have demonstrated applications of sensors networks (Caubel et al., 2019; Gao et al., 2015; Jiao et al., 2016; Popoola et al., 2018; Yuval et al., 2019). Distributed sensing of air quality can be seen as an important progression towards more comprehensive understanding of city-scale air quality dynamics as air pollution, and especially particulate matter (PM), may have highly



localized concentration "hotspots" in urban areas. Practical limitations, such as expensiveness and bulkiness, constrains the use of conventional instrumentation in monitoring networks and therefore low-cost sensors could have an essential role in

the spatial extension of measurement coverage.

Numerous field studies have been conducted previously, and majority of these have underlined the potential usefulness of optical particulate matter sensors (Karagulian et al., 2019; Rai et al., 2017). In spite of this, it has also been emphasized that the risk of sensor misuse is still high, and that some external factors, such as relative humidity, may produce significant

measurement artefacts to the data (Jayaratne et al., 2018; Kuula et al., 2018; Liu et al., 2019). In comparison to gas sensing, PM measurements are notably more challenging considering that ambient particle sizes, and their respective size distributions, may vary significantly from source to source and from location to location, and along with size, particle physical properties such as shape and refractive index also have an effect on the sensor output. Several studies have pointed out that along with dynamic adjustment for meteorological parameters, on-site calibrations are required in order to achieve

higher levels of accuracy and precision (Zheng et al., 2018). However, when considering advanced calibration techniques, Schneider et al. (2019) has raised a valid point noting that it may be unclear whether the sensor data resulting from complex correction and conversion processes (e.g. machine learning) is still a legitimate and independent product of the sensor measurement and not a combination of secondary data and statistical model prediction. This is an important remark when evaluating the usability of sensors, and moreover, highlights the need to identify the underlying reasons causing inaccuracies

in low-cost sensor measurements.

While field evaluations are a natural step towards understanding and developing sensors, they provide limited information about the detailed sensor response characteristics. In particular, less attention has been paid to the investigation of particle size-selectivity of sensors. Although few studies have noted that the detectable particle size ranges of sensors may be

significantly different from the ones declared in their respective technical specification sheets (Budde et al., 2018; Levy Zamora et al., 2019), it is not a commonly considered factor when assessing sensor accuracy. Thus more research is needed. The objective of this study was to investigate and characterize the size-selectiveness of some of the optical low-cost sensors commonly appearing in the literature. The evaluated sensors were Plantower PMS5003, Nova SDS011, Sensirion SPS30, Sharp GP2Y1010AU0F, Shinyei PPD42NS, and Omron B5W-ld0101. Along with these low-cost sensors, a mid-cost optical

aerosol spectrometer (GRIMM model 1.108, GRIMM Aerosol GmbH., Germany) was evaluated cursory to highlight the differences between responses of the low-cost and mid-cost devices. The investigation of size-selectivity was carried out in laboratory using a novel reference aerosol generation system capable of steadily producing monodisperse particles of different sizes. Sensor responses were compared to a reference instrument (APS, Aerodynamic Particle Sizer 3321, TSI Inc., USA), and detectable particle size ranges of the sensors were obtained.



## 2 Methods

### 2.1 Evaluated sensors

The sensors evaluated in this study, and their main detection properties, are listed in Table 1. The optical detection configurations of these sensors are arranged in either 90 or 120 degree scattering angle, and a red laser or an infrared (IR) light emitting diode (LED) is used as a light source. Sensors utilizing an LED are equipped with additional light focusing lenses. The optical chamber itself is constructed of an injection molded plastic body which is placed onto an electronic circuit board. The PMS5003, SDS011, and SPS30 use fans to generate sample flow whereas the PPD42 and B5W utilize natural convection resulting from a heating resistor. Sampling of the GP2Y1010AU0F is based on diffusion. The optical configurations and plastic body layouts are shown in Supplemental Figure S1.

All sensor units were in original condition except the PPD42 and B5W sensors which had their air heating resistors removed. The evaluation platform used in this study did not necessarily require independent means of sample flow. Furthermore, holes were drilled to the plastic body of the PPD42 in order to ensure that the sample aerosol could reach to the optical detection volume. The inlet of the PPD42 was originally designed to be on top of the plastic body (facing towards the electronic circuit board), and therefore, when the electronic circuit board of the sensor was orientated in parallel with the sample stream, majority of the particles would have bypassed the sensor. In general, along with the PPD42, the plastic body layouts of the PMS5003 and SPS30 are susceptible to inertial deposition losses due to their 90 degree elbows in particle stream pathways. However, the sub-optimal layouts of these sensors are better compromised by the more stable sample flow system (i.e. fan instead of convection).

The PMS5003, SDS011, and SPS30 sensors have digital outputs whereas the others are analogue based. Along with the sensor outputs shown in Table 1, the PMS5003 and SPS30 sensors also output particle number concentrations, but these signals were not used as the response comparison to the reference instrument was carried out using only mass concertation values. This decision was based on the observation that low-cost sensors have been predominantly used to measure mass concentration and not number concentration. Three units for each sensor model were evaluated in order to assess their inter-unit variation. The mid-cost GRIMM 1.108, which was used here for demonstration purposes, is an optical aerosol spectrometer with 15 size bins (from 0.23 to 20 μm). Previous evaluations of the GRIMM 1.108 has shown its response to be similar to the APS (Peters et al., 2006), and furthermore, its accuracy being comparable to other mass measurement methods, such as the filter weighing method (Burkart et al., 2010).



## 2.2 Reference aerosol

### 95  2.2.1 Vibrating orifice aerosol generator and gradient elution pump

The operating principle of a Vibrating Orifice Aerosol Generator 3450 (VOAG, TSI Inc., USA) is based on the instability and break-up of a cylindrical liquid jet. Mechanical disturbances of a resonance frequency vibration disintegrates the cylindrical jet into uniform droplets, which are dispersed into an aerosol flow system with appropriate dilution air. Dispersed droplets evaporate before significant coagulation occurs, and form particles from the non-volatile solute dissolved in the

volatile liquid. If the droplet liquid is non-volatile, the particle diameter and droplet diameter are equal. Otherwise, the produced particle size is calculable from the volumetric fraction of the non-volatile solute, as shown in Eqs. 1-2. This aerosol generation method and an apparatus (predecessor of the TSI's VOAG) was first introduced by Berglund and Liu in 1973.

$$D_d = \left(\frac{6Q}{\pi f}\right)^{1/3} \tag{1}$$


Where $D_d$ is the generated droplet diameter, $Q$ is the solution feed rate, and $f$ is the disturbance frequency.

$$D_p = (C + I)^{1/3} * D_d \tag{2}$$

Where $D_p$ is the diameter of the formed particle, $C$ is the volumetric concentration of the non-volatile solute in the volatile liquid (typically 2-propanol or purified water), and $I$ is the volumetric fraction of impurity in the volatile liquid.

The output aerosol number concentration of the VOAG has relative standard deviation of less than 3 %, and the formed particle size distribution is monodisperse having geometric standard deviation (GSD) less than 1.014 (Berglund and Liu,

1973). These, and particularly the capability to produce highly monodisperse size distribution, are important features regarding sensor size-selectivity evaluation; while polydisperse aerosol can be used to estimate e.g. response stability and linearity to varying concentration levels (Hapidin et al., 2019; Papapostolou et al., 2017; Sayahi et al., 2019a), the presence of multiple different sized particles prevents the distinction between sensor response and specific particle size. The greatest deficiency of the VOAG (and the main limitation of this study) is that its smallest producible particle size is limited by the

impurity within the carrier liquid, and is in practice limited to approximately 0.55 µm.

The novelty of the aerosol generation method used in this research is based on the observation that blending of two liquid solutions with different non-volatile concentrations produces a stable particle size gradient, respective of the concentrations of the blending solutions. In other words, the produced particle size of the monodisperse and constant number concentration

reference aerosol can be controlled by feeding solutions of different non-volatile concentrations to the VOAG, one after each



other. Such aerosol generation technique was first utilized by Kuula et al. (2017) who accomplished the solution blending with a supplementary syringe pump and manually operated 3-way valve. In this study, however, the solution feeding was done with a gradient elution pump used typically in ion-chromatography (GP50, Dionex Inc., USA). The GP50 gradient pump has four different eluent channels, and is capable of dispensing liquids with high pressure (max. 5000 psi) and accurate

volume flow rate (0.04 – 10.0 mL min-1 in increments of 0.01 mL min-1). The four eluent channels can be mixed with a resolution of 0.1 % (combined output of the four channels always 100 %), and, moreover, the GP50 has a user-interface which enables the operator to generate parameterized eluent dispensing programs. In essence, the utilization of the GP50 allows the user to freely choose and produce monodisperse aerosols of desired particle sizes without the tuning of VOAG running parameters or manual alternation of the liquid concentrations. Furthermore, the preconfigured dispensing programs

are fully automated making the comparison of consecutive test runs more reliable.

### 2.2.2 Sampling configuration

A schematic figure of the used test setup is shown in Figure 1. Reference aerosol was generated using the VOAG-GP50-system as described in the previous section. Dioctyl sebacate (DOS, density of 0.914 g cm-3) was used as a non-volatile solute in a 2-propanol solvent (> 99.999 %, Sigma-Aldrich), and the formed particles were thereby transparent oil droplets.

Running parameters of the VOAG and GP50 are shown in Supplemental Table S1. The three different DOS concentrations (A–C) refers to the four different eluent channels of the GP50 (use of three channels was sufficient for this study).

The GP50 was operated in a Method-mode meaning that an automated program was used to dispense the liquids. A program (i.e. method) constitutes of consecutive time steps in which the blending ratios of eluent channels, step durations, and flow

rate can be defined separately for each time step. Executing the program means that the GP50 dispenses the liquids according to the settings determined in each step. The program used in this evaluation consisted of 10 steps in which the produced particle sizes were logarithmically distributed from 0.45 to 9.78 µm. The calculated blending ratios and respective particle sizes are shown in Supplemental Table S2. Step duration of 5 min was used and thus a single test run lasted approximately 60 minutes. Dead volumes in the GP50 and VOAG slightly extend the theoretical run time duration. A

complete test run can be performed as quickly as in 15 minutes but this results in fewer measurement points and weaker statistical power. An example of the produced reference aerosol number size distribution measured with the APS is shown in Figure 2.

Formed particles were neutralized in the dispersion outlet of the VOAG, and further fed into a flow splitting section where

the reference aerosol was directed symmetrically to both reference instrument (Aerodynamic Particle Sizer 3321, TSI Inc., USA) and sensor. The sensors were encapsulated in 3d printed air-tight enclosures with an external pump connected to it in order to ensure appropriate sample flow through the sensor. The symmetrical sample flow rate was set to be 1 L min-1 as this was the aerosol flow rate of the APS (sheath flow of the APS taken from the laboratory air). For the PMS5003 and



SPS30 sensors, an exhaust deflector was used to prevent unwanted sample mixing resulting from the fan outlet, which for
these sensors, was situated right next to the sensor inlet. A schematic figure of the sampling arrangements is shown in
Supplemental Figure S2.

**2.3 Data processing**

The output signal of the evaluated sensor and APS was measured synchronously using a 10 second time resolution and
moving average. Any raw measurement point which had GSD (calculated from the APS data) exceeding 1.2 was
disregarded, but typically the GSD values were within 1.04 – 1.08 range. The sensor bias was set to zero by sampling clean
air for 10 minutes (60 data points) and then subtracting the clean air response from the test aerosol response. The bias
correction was only relevant for the GP2Y1010AU0f and B5W sensors. In order to prevent arbitrary unit comparisons, the
sensor response was normalized using Eq. 3:

$$Normalized\ detection\ efficiency = \frac{\frac{Sensor_i}{APS_i}}{\max(\frac{Sensor}{APS})}$$  (3)

Where $i$ is the ith measurement point, $Sensor$ is the sensor signal, and $APS$ is the APS total mass concentration.

The normalized 10 second resolution data was divided into 30 logarithmically distributed size bins (from 0.45 to 9.73 µm)
according to the count median diameters (CMD, aerodynamic) measured by the APS. An average sensor response as a
function of average CMD was then calculated for each size bin. The decision to divide the data into 30 bins was based on the
clarity of the produced figures and statistically sufficient number of measurement points belonging to each bin. This process
was completed for three different sensor units, and a combined (average and standard deviation) sensor response was
calculated. Valid detection ranges, which were defined as the upper half of the detection efficiency curve, of the sensors
were linearly interpolated from the average response functions. The size bins of PMS5003, SPS30, SDS011, and B5W were
discretized so that no overlapping signals were obtained. For example, the outputs of the SDS011 were used as PM2.5 and
PM10-2.5 (PM10-2.5 calculated as PM10 - PM2.5) instead of PM2.5 and PM10.

**3 Results and discussion**

**3.1 GRIMM model 1.108**

The response curves of the GRIMM 1.108 are shown in Figure 3. For the sake of clarity, the degrees of measurement
variation have been excluded from the figure. The previously described data processing technique was used, and the
comparison to APS was conducted with mass concentration. Bins 14 and 15, which correspond to 10 – 15 and 15 – 20 µm,
respectively, are not shown here as they did not produce any signal (as expected). The normalized detection efficiency of 70



– 90 % results from the average efficiency from multiple data points and, in this case, does not imply that the GRIMM would
systematically underestimate particle mass concentrations. The same applies for respective sensor response figures (next section).

The response characteristics of the GRIMM are in line with its technical specifications showing that each size bin only corresponds to its specific detection range. A flat response curve would indicate that the strength of the output signal remains
unchanged regardless of the particle size, and thus would show that the size bin is unable to make distinction between different particle sizes. Some mismatch between the particle sizing of the APS and GRIMM can be observed as a result of different particle sizing techniques (time of flight and optical), but this is trivial considering the objective of this study. The purpose of this figure is to highlight how an aerosol measurement device with several particle sizing bins should respond to the evaluation method used in this study.

**3.2 Low-cost sensors**

Response functions of the evaluated sensors are shown in Figures 4a-f. The coloured circles represent the calculated average responses of the three sensor units and the shaded background areas respective standard deviations. Standard deviations of the average CMDs were negligible due to the reliable and reproducible test method. Figure legends correspond to the bin size ranges stated by the corresponding manufacturer.


**Plantower PMS5003**

According to Figure 4a, it is apparent that the particle sizing of the different bins is not working properly. The first and second bin (supposedly corresponding to 0.3 – 1.0 and 1.0 – 2.5 μm) are similar having valid detection ranges of approximately < 0.7 μm and < 0.8 μm, respectively (valid detection ranges were defined as the upper half of the detection
range, see section Data processing). It is possible that the lower cut-points of these bins reaches close to 0.3 μm, as stated by the manufacturer, but this could not be confirmed using the VOAG-GP50 system. The third bin is noisier as indicated by the larger standard deviations, and is significantly off of its stated detection range (2.5 – 10 μm).

Based on the test, the PMS5003 cannot be used to measure coarse mode particles (2.5 – 10 μm), and furthermore, its ability
to measure PM2.5 is dependent on the stability of the ambient air size distribution; if the proportions of mass in e.g. < 0.8 and > 0.8 μm fractions changes significantly, the PMS5003 is susceptible to inaccuracies. This is because its valid detection range cannot account for changes occurring in parts of the size distribution which it essentially cannot observe. However, if the ambient size distribution is stable, the PMS5003 can be adjusted to measure PM2.5 with reasonable accuracy (Bulot et al., 2019; Feenstra et al., 2019; Magi et al., 2019; Malings et al., 2019). Similarly, the validity of PM10 measurements can be
ensured only when the proportion of mass in > 0.7 or > 0.8 μm size fraction is either constant or negligible with respect to the total PM10 mass. In reality, this is rarely the case and therefore a high risk of sensor misuse is posed. This observation is



in line with the findings of previous studies (Laquai, 2017b; Levy Zamora et al., 2019; Li et al., 2019; Sayahi et al., 2019b) which showed, for example, that the PMS5003 could not detect a substantial dust storm episode while deployed in field. The most accurate and reliable results are most likely achieved for the PM1 size fraction by using either bin 1 or bin 2 signals.


**Nova SDS011**

Response function of the SDS011 is shown in Figure 4b. Contrary to the PMS5003, the SDS011 exhibits two more clear different detection ranges; the first bin (0.3 – 2.5 µm) corresponds approximately to < 0.8 µm and the second bin (2.5 – 10 µm) corresponds approximately to 0.7 – 1.7 µm. Similarly to the PMS5003, the SDS011 is not suitable for the measurement

of coarse mode particles, and the measurements of PM10 can be grossly inaccurate. Previous studies have also noted this (Budde et al., 2018; Laquai, 2017a). However, due to the clearer difference between the two detection ranges, the SDS011 has potential to measure PM2.5 more accurately than the PMS5003. Moreover, an artificial correction factor approximating changes in the measured size distribution may be possible to calculate from the ratios of bin 1 and bin 2 signal strengths. Previous studies have shown that the SDS011 can be reasonably accurate in the measurements of PM2.5 (Badura et al.,

2018; Liu et al., 2019).

**Sensirion SPS30**

Response function of the SPS30 is shown in Figure 4c. The valid detection range of the first bin (0.3 – 1.0 µm) is approximately < 0.9 µm. The second, third, and fourth bin (supposedly corresponding to 1.0 – 2.5, 2.5 – 4.0, and 4.0 – 10

µm) are nearly identical having valid detection ranges of approximately 0.7 – 1.3 µm. The identical detection ranges indicates that these bins may have been factory calibrated using the same test aerosol. The SPS30 is a relatively new sensor (introduced to the markets in late 2018), and Web of Science nor Scopus literature research showed any existing studies as of September 2019. However, South Coast Air Quality Management District (SCAQMD) has conducted a preliminary field test where three SPS30 units were compared to three different federal equivalent method (FEM) monitors (SCAQMD, 2019).

The results of this test showed that the SPS30 sensors had very low cross-unit variability (~1, 1.3, and 2.4 % for PM1, PM2.5, and PM10, respectively) and, more importantly, the accuracies for the measurement of PM1, PM2.5 and PM10 decreased from R2 ~ 0.91 to 0.83 and further down to 0.12, respectively. These observations are in high agreement with the results of this study, and furthermore, illustrate how a sensor with limited operational range may exhibit a near regulatory grade accuracy if the measured size fraction is in align with the valid detection range of the sensor (< 0.9 µm and PM1). On

the other hand, the severity of data misinterpretation is apparent when the sensor measurement is extended to cover particle sizes which it cannot observe.

**Sharp GP2Y1010AU0F**

Response function of the GP2Y1010AU0F is shown in Figure 4d, and its valid detection range appears to be approximately

1.8 – 6.5 µm. This is problematic considering the typically monitored PM2.5 and PM10 parameters as the sensor reacts to



particulate mass belonging to both < 2.5 and > 2.5 μm size fractions. Consequently, the measurement output can be particularly difficult to interpret if this detail is not known. Nevertheless, the GP2Y1010AU0F has been used in variety of different applications (Alvarado et al., 2015; Zuidema et al., 2019).

260 Several laboratory evaluations have been conducted previously for the GP2Y1010AU0F, but none of these have assessed its detection range using monodisperse test aerosols (Li and Biswas, 2017; Manikonda et al., 2016; Sousan et al., 2016). A study of Wang et al. (2015) used atomized polystyrene latex (PSL) particles to evaluate the effect of particle size to the GP2Y1010AU0F response, but no concluding remarks can be obtained from these results (Wang et al., 2015). The study method utilized only three different sized PSLs, and moreover, was not designed to investigate the complete detection range

265 of the GP2Y1010AU0F to begin with. However, some of the results implied that the sensor could detect particles as small as 0.3 μm, which is in significant conflict with the results of this study. There is no obvious explanation for this.

**Shinyei PPD42**

Response functions of the three PPD42 sensor units are shown in Figure 4e. Contrary to the other sensors, a combined

270 response function was not calculated as the three units exhibited significantly different response characteristics. The circles and shaded background areas in this case represent average responses and respective standard deviations of the individual sensor units (calculated from the ~ 300 raw data points). The valid detection range of the first unit is 1.0 – 2.1 μm, and it is likely to be best suited for PM2.5 measurements. However, the low detection efficiency of < 1.0 μm sized particles may hinder its accuracy considerably. Valid detection ranges of the second and third unit are > 5.9 and 1.5 – 4.9 μm indicating

275 preferable applicability to coarse mode particle measurements. Previous laboratory evaluations have noted that the PPD42 output is a function of particle size but could not provide more detailed analysis of the complete detection range (Austin et al., 2015; Wang et al., 2015). A study of Kuula et al. (2017) reported a valid detection range of approximately 2.5 – 4.0 μm, which is in the same range as the third unit of this study (Kuula et al., 2017).

280 Due to the apparent inter-unit inconsistency in valid detection ranges, it is evident that the response characteristics of the PPD42 have to be quantified case-by-case before reliable measurements can be achieved. Accordingly, the inconsistent response characteristics may also contribute to the notion that previous field evaluation studies have achieved varying results regarding the accuracy of PPD42; studies of Bai et al. (2019) and Holstius et al. (2014) reported R2 values of 0.75 and 0.55 – 0.60, respectively, for the measurement of PM2.5, whereas studies of N. E. Johnson et al. (2018) and K. K. Johnson et al.

285 (2018) reported more modest values of 0.36 – 0.51 and 0 – 0.28, respectively (Bai et al., 2019; Holstius et al., 2014; Johnson et al., 2018a, 2018b). On the other hand, studies of Kuula et al. (2017, 2018) showed that higher levels of accuracy can be achieved if the measured size fraction is targeted to correspond the characteristic response function of the PPD42 (R2 = 0.96 and R2 = 0.87, respectively) (Kuula et al., 2017, 2018).



**Omron B5W**

Response function of the B5W is shown in Figure 4f. The two size bins exhibit two clearly different detection ranges (0.6 – 1.0 and > 3.2 µm, respectively) which are reasonably close to the ones declared by the manufacturer (0.5 – 2.5 and > 2.5 µm, respectively). In fact, out of all sensors, the B5W appears to be the most prominent sensor for ambient monitoring of PM2.5 and PM10-2.5 size fractions. In comparison to e.g. SDS011 and SPS30, the usability of the B5W may be hindered by its temperature gradient based sampling method as it is not as reliable as the respective fan based method. Nonetheless, it is the only sensor capable of measuring both fine and coarse fraction particles. Web of Science nor Scopus literature review showed existing studies for the Omron B5W.

**Conclusions**

According to the results obtained in this study, optical low-cost sensors exhibit widely varying response characteristics regarding their size-selectivity (from < 0.7 to > 5.9 µm, Table 2). However, none of the sensors had exactly the same response characteristics as stated by their manufacturers. This provides some insight and evidence to the notion that particle size-selectivity may have an essential role in the error source analysis of sensors, and furthermore, underlines that scientists, as well as manufacturers for that matter, need to acknowledge limitations related to this. Respectively, it is worth noting that attempts to artificially extend the operational range of sensors beyond their practical capabilities using complex statistical models can be unreasonable and may lead to misleading conclusions. Empirical corrections for known artefacts, such as the humidity, can be justifiable, however, in general sensor data and advanced modelling techniques should be merged cautiously in order to pertain the validity and representativeness of the data.

Cursory comparison to a mid-cost aerosol spectrometer (GRIMM 1.108) shows that low-cost sensor development is still considerably behind its more expensive alternative; while the GRIMM 1.108 could sufficiently characterize particle sizes with up to 15 different size bins, the low-cost sensors could only achieve independent responses for 1-2 to bins. This is a major weakness considering that the ability to correctly measure particle size is at the foundation of accurate mass measurement (mass α dp3). Development of low-cost sensors should focus on increasing the number size bins, and more importantly, making sure that each size bin is calibrated correctly. Improperly configured bin sizing poses a significant risk of data misinterpretation, and will inevitably lead to inaccurate measurements. Low number of size bins limits the valid operational range of sensors, however, it is unclear how the amount of advanced measurement features and low unit cost should be reconciled.

The VOAG-GP50 aerosol generation system described in this study, introduced a novel approach in how aerosol measurement devices can be evaluated quickly and efficiently. The use of GP50 gradient pump eliminates much of the manual labour which was previously inseparable part of the VOAG operation, and thus makes the generation of reference



aerosols more consistent and reliable. Its automated dispensing programs allows for highly repeatable testing, and furthermore, the four different eluent channels enables the operator to freely pick and choose desired particle size to be produced. Along with saving manual labour and time, this is also a cost-saving feature as traditionally used polystyrene latex

(PSL) particle are not needed. Considering these matters, the VOAG-GP50 system can potentially be scaled to industrial level operation which is an intriguing feature when considering mass deployment of sensors and their respective quality assurance and control.

**Author contribution**

JK and TM designed the experimental setup, and JK carried out the tests. KT had an important role in refurbishing the

gradient elution pump. SM and OG provided some of the sensors. JK was responsible for the data analysis, although all co-authors provided valuable feedback, particularly TM. JK wrote the manuscript with the help of all co-authors.

**Conflict of interest**

The authors declare no competing financial interest.

**Acknowledgements**

This study was funded by the Urban innovative actions initiative of the European Regional Development Fund (project HOPE; Healthy Outdoor Premises for Everyone, project no: UIA03-240), and by the European Union's Horizon 2020 research and innovation programme under grant agreement No 689954 (project iSCAPE; Improving the Smart Control of Air Pollution in Europe).

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





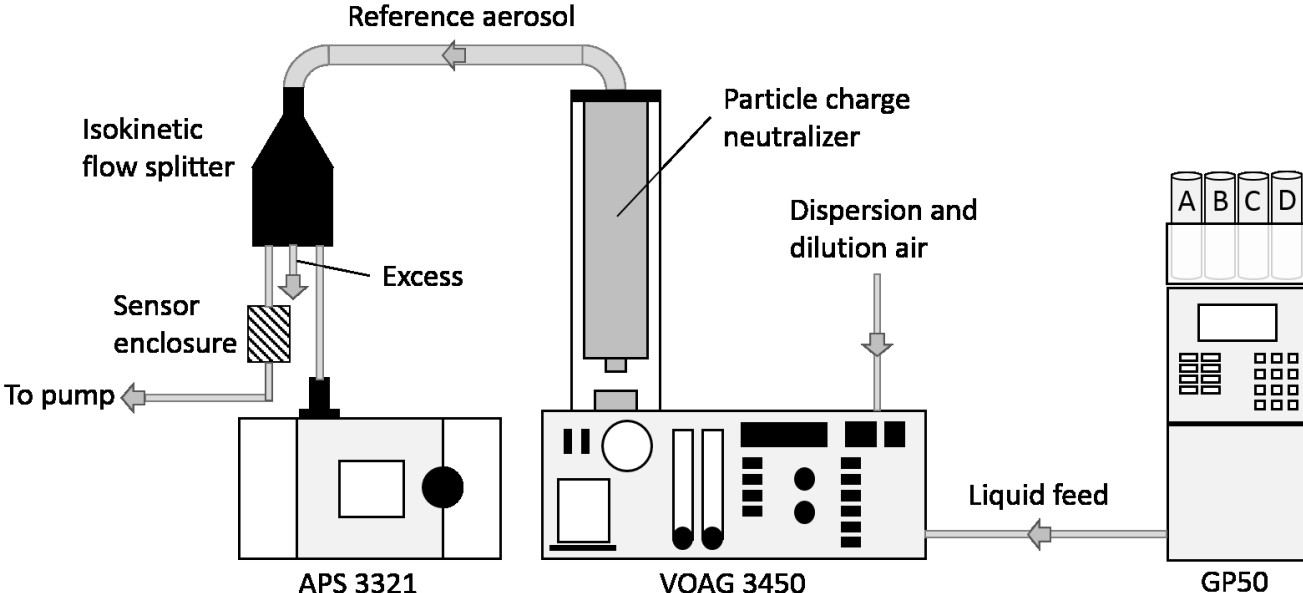

**Figure 1: Schematic of the sensor evaluation setup.**





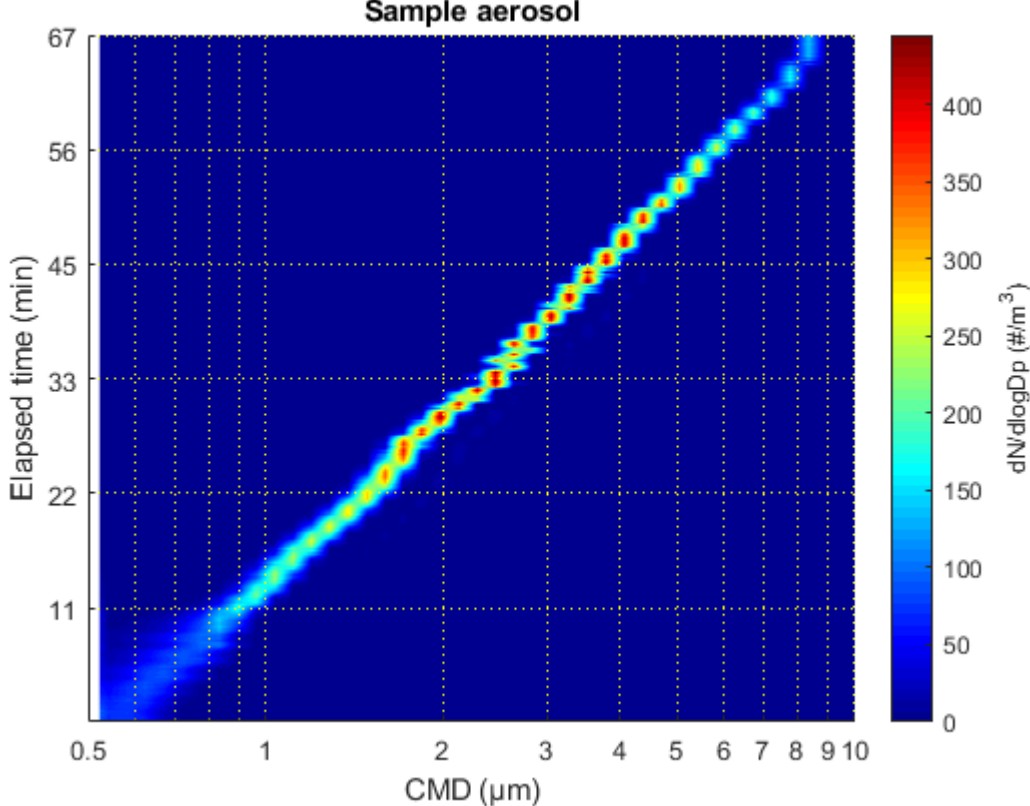

**Figure 2. An example of a produced reference aerosol. Decreasing number concentrations below 1 µm and above 5 µm result from the approaching lower detection limit (0.5 µm) of the APS and from the increasing inertial deposition losses in the sampling lines, respectively. This had, however, no effect on the evaluation results as the sensor response was normalized against the concentration measured by the APS. The GSD of the size distribution remains below 1.2.**






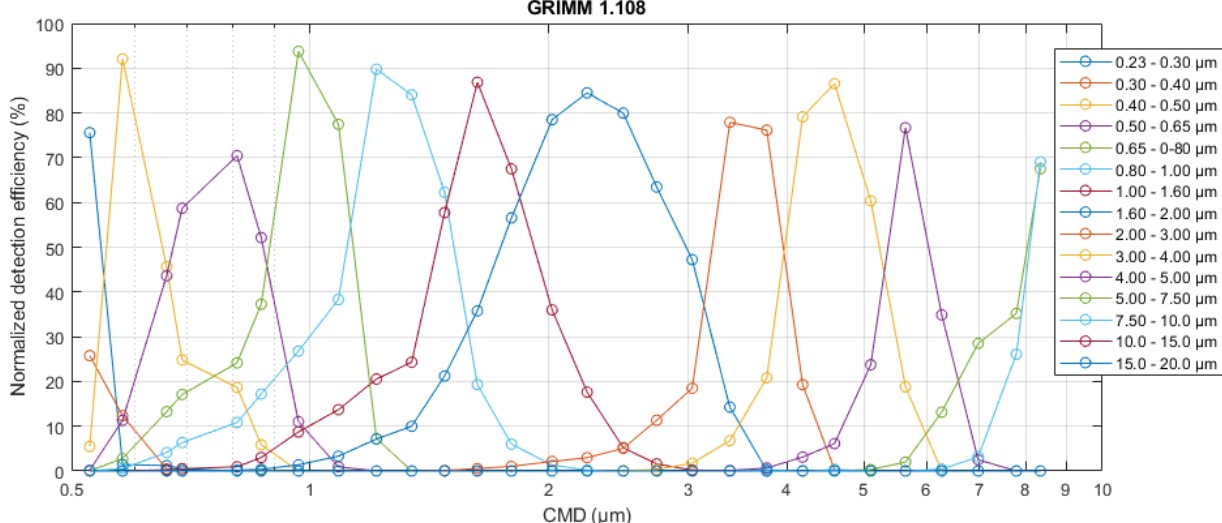

**Figure 3. Illustration of the response characteristics of the GRIMM 1.108 aerosol spectrometer. Consecutively increasing and decreasing response curves indicates that the particle sizing of the instrument is functioning properly.**






**Figure 4. Measured particle size response functions of the low-cost sensors.**





**Table 1. Basic features of the evaluated sensors declared by the manufacturers.**

| Low-cost sensor | Detectable size range (μm) | Number of size bins | Scattering angle | Wavelength | Sensor output |
|---|---|---|---|---|---|
| Plantower PMS5003 | 0.3 – 10 | 3 | 90° | Red (laser) | $PM_1$, $PM_{2.5}$, $PM_{10}$ |
| Nova SDS011 | 0.3 – 10 | 2 | 90° | Red (laser) | $PM_{2.5}$, $PM_{10}$ |
| Sensirion SPS30 | 0.3 – 10 | 4 | 90° | Red (laser) | $PM_1$, $PM_{2.5}$, $PM_4$, $PM_{10}$ |
| Sharp GP2Y1010AU0F | n/a | 1 | 120° | IR (LED) | Voltage level |
| Shinyei PPD42 | > 1 | 1 | 120° | IR (LED) | PWM-signal |
| Omron B5W-ld0101[*] | > 0.5 | 2 | 120° | IR (LED) | Pulse count (> 0.5, > 2.5 μm) |
| Mid-cost monitor: GRIMM 1.108 | 0.23 – 20 | 15 | 90° | 780 nm (laser) | Number, surface, and mass conc. |

[*] Manually adjusted threshold voltage was set to 0.5V as recommended by the manufacturer.



**Table 2. Valid detection ranges of the evaluated sensors. Symbols of "greater than" or "smaller than" refers to cases where the other end of the size cut-point was outside of the particle size range producible by the VOAG-GP50 system (0.45 – 9.73 µm). Units are in µm.**

| Sensor | Bin 1 | Bin 2 | Bin 3 | Bin 4 |
|---|---|---|---|---|
| Plantower PMS5003 | < 0.7 | < 0.8 | < 1.0 (noisy) | - |
| Nova SDS011 | < 0.8 | 0.7 – 1.7 | - | - |
| Sensirion SPS30 | < 0.9 | 0.7 – 1.3 | 0.7 – 1.3 | 0.7 – 1.3 |
| Sharp GP2Y1010AU0F | 1.8 – 6.5 | - | - | - |
| Shinyei PPD42* | 1.0 – 2.1 | > 5.9 | 1.5 – 4.9 | - |
| Omron B5W | 0.6 – 1.0 | > 3.2 | - | - |

* Valid detection ranges of the individual sensors, not bins.