# Peer review of "Laboratory evaluation of particle-size selectivity of optical low-cost particulate matter sensors"

_Atmospheric Measurement Techniques, 2019_

## Referee Comment (RC1) · Anonymous Referee #2 · 27 Dec 2019

General Comments This manuscript describes the development of a system to provide a flow of monodisperse particles, and the authors then use this system to evaluate several common low-cost sensors. The description and evaluation of the new system (VOAG-GP50) are clearly described. Although their conclusion that low-cost sensors mis-classify particle sizes is not new, the systematic evaluation of size selectivity in low-cost sensors is a valuable contribution to the field. However, the paper would be stronger if the authors improved the clarity of the data-processing section, address a few questions regarding sample flow rate, and polish the language.

Specific comments:

The data processing section needs some clarification. The authors discuss dividing the data into 30 size bins, but in line 146 they discuss 10 steps to produce different

particle sizes. The authors need to clarify how 10 steps can yield 30 different size bins. In Lines 175 to 180, the authors base their discussion of valid detection ranges on a detection efficiency curve, but I could not find a discussion of how they define a detection efficiency curve. I would suggest providing an example in the supplementary material and illustrating how the upper $\frac{1}{2}$ of this curve is defined.

I have some concerns regarding the effect of sample flowrate on the low-cost sensors. In the experimental setup, a pump draws the monodisperse particles into the sensor housing at a flow rate of 1 lpm. Figure S2 shows the sensor housing and placement of the low-cost sensors with the flow directed at the sensor inlet (mostly). The authors should consider whether this setup may be skewing their results. This is particularly important for sensors with fans that are designed to operate at a specific flowrate. It is possible that pushing a flowrate that differs from the design flow rate could alter the results. For example the PMS sensor has a volumetric flowrate of approximately 0.1 lpm (which is 10x lower than the volumetric flowrate into the sensor housing). Granted not all of the 1 lpm would flow into the sensor, but this is worth considering.

The manuscript needs a thorough review and edit by a native English speaker. The language is awkward and sometimes confusing. I am including a few examples from the abstract, but this list is not comprehensive: - "due to their prostective nature regarding spatial extension of measurement coverage". Vague and awkward wording. - "sensors can be useful in achieving this goal". No goal is mentioned previously. - "it is often reminded that the risk of sensor misuse". Improper usage.

Technical corrections:

Line 121. Do the authors mean stable particle size distribution rather than particle size gradient? If they mean particle size gradient, this needs to be explained.

Line 185. The authors should clarify what the response curves are for. The 10-step (30-bin) generation of monodisperse particle sizes?

Line 203. The authors should provide the standard deviations of the CMDs. In Figure 2, the text says "The GSD of the size distribution remains below 1.2, but line 113 says that the VOAG has relative standard deviation of less than 3 %."

The authors mention the size limitation of the APS as being a limiting factor in the analysis (Figure 2), but in line 119 they mention that the VOAG cannot reliably generate particles smaller than 0.55 um. This limitation should be mentioned in Figure 2 in addition to the APS limitation. It would also be worth mentioning this important limitation in the abstract.

Figure 1 – The GRIMM is not shown. Where does the GRIMM draw its sample? How are the flows distributed symmetrically between the APS and GRIMM since the GRIMM's flow is 1.2 lpm whereas the GRIMM is 1 lpm?

---

## Referee Comment (RC2) · Anonymous Referee #1 · 3 Feb 2020

In this work, Kuula et al. present an investigation of counting efficiency as a function of size for a series of low-cost light scattering devices. The research question that the authors seek to answer (whether low-cost sensors can correctly assign particles to the appropriate PM mass fractions) is an important one. The work appears well thought out and carefully conducted, although the manuscript is below average from an English language perspective. There are dozens of grammatical and stylistic mistakes that will need attention prior to publication. However, the authors present a novel experimental procedure and some interesting results, so I am supportive of publication if this and other concerns are addressed in a revised manuscript. Comments on how the presentation of the methods and the interpretation of the results could be improved are below.

[Figure]

Major Comments

One major drawback of this work is that the authors appear to have used a "forced flow" of 1 L/min through each sensor, which may not be in accordance with the manufacturer's intended use. The use of a non-standard flow rate (i.e., one that is different from the manufacturer's recommendation) could compromise a sensor in a number of ways. For example, it could lead to differential inertial losses of particles as they transmit through the sensor. Or, it could lead to variable pulse widths of particles during detection that deviate from the expected pulse widths during calibration. The authors need to verify that this method of testing under a forced flowrate did not affect sensor response. One way would be to repeat these tests at flow rates of 0.5 and 2 Lpm to see if the response factors change. If they do change, then the results presented here might lead to misinterpretation...I do not support publication of this manuscript until this issue is investigated further and/or resolved.

I think the VOAG method for generating a sequence of monodisperse particles of varying size is both novel and useful to the community. However, I would appreciate more detail on how the method is implemented so that others can properly reproduce this method. Such details could be enumerated as supplemental material.

Comments on the Methods section

The Methods section could be organized better and was missing key information needed to reproduce the work.

Comments on technical issues:

1. Table 1: The "number of size bins" column is problematic. It's not appropriate to compare the number of PM mass fractions reported by the low-cost sensors to the number of size bins that the GRIMM 1.108 classifies particle into. Perhaps the authors could remedy this problem by creating two columns: one labeled "number of mass fractions reported" and "number of particle size bins". For example, the SPS30 reports

data in 4 mass fractions and 5 particle size bins.

2. Line 76: Why did the authors remove the mechanism that drives sample flow through some of the sensors but not others? Specifically, why did they remove the air heating resistors from the PPD42 and B5W but not remove the fans from the PMS5003, SDS011, and SPS30? This discrepancy seems like it could lead to an unfair comparison between different sensors. A sensor's output is partially determined by how quickly and efficiently particles are drawn into the sensing zone. If the flow rate through the sensor is modified, the output of the modified sensor might differ from the output of the "as-purchased" sensor. For example, results presented by Tryner et al. (doi: 10.1039/C9EM00234K) indicate that varying the flow rate through a modified PMS5003 sensor changes the output. Since the authors modified some of the sensors used in this study, the results presented here might not be directly transferrable to unmodified sensors being used in the field.

3. Lines 85-89: The PMS5003 reports each mass fraction two ways: "CF=1, standard particle" and "under atmospheric environment". Which values were used in the data analyses presented here? 4. Lines 164-165: Please state the fraction of raw measurement points that were disregarded.

5. Line 170: Equation 3 is not clear. What is the maximum Sensor/APS ratio? Was this the maximum ratio measured during the entire 60-minute long test run?

6. Line 175: One issue is that the APS is known to have poor counting efficiency for liquid droplets above about 5 microns in diameter. Did the authors account for this artifact? See https://doi.org/10.1016/j.jaerosci.2005.03.009

7. Line 179-180: This definition of "valid detection ranges" is not clear. Do the authors mean that the valid detection range was the range of CMDs for which the normalized detection efficiency was greater than 50%? If so, please consider also rephrasing the text on lines 209-210 as something like "(the valid detection range was defined as the range of particle sizes for which normalized detection efficiency was > 50%)".

Comments on organization:

1. Lines 75-84: This information should be moved to the "Sampling configuration" section. These methods are difficult to understand or justify without additional knowledge of the system used to pass monodisperse aerosol through the sensor.

2. Lines 85-82: This information should be moved to the "Data processing" section.

3. Lines 89-90: Please move the sentence that starts "Three units of each sensor…" to the end of the first paragraph in Section 2.1.

4. Lines 90-93: Please start a new paragraph here that contains the information on the GRIMM 1.108.

5. Line 95: Section 2.2.1 should start with a sentence that says, "The aerosol sampled by the low-cost sensors was generated using a Vibrating Orifice Aerosol Generator 3450 (VOAG, TSI Inc., USA)." That way readers will know why the information contained in this section is relevant.

6. Lines 101-102, "This aerosol generation method…": This sentence is neither relevant nor helpful. Please delete it.

7. Lines 121-125: The second sentence in this paragraph is more helpful than the first. I suggest rephrasing as "The novelty of the aerosol generation method used in this research is based on the observation that the particle size of the monodisperse and constant number concentration reference aerosol can be controlled by feeding solutions with different non-volatile concentrations to the VOAG, one after each other."

8. Lines 161-162: Please show at least one example panel from Figure S2 in the main text.

9. Lines 176-177, "The decision to divide the data into 30 bins…": This sentence is not necessary. Please consider deleting it.

10. Lines 186-187: The sentence that begins "The previously-described data processing technique. . ." should be moved to the "Data processing" section.

Comments on the Results section

Comments on technical issues:

1. Overall, the authors interpretation of their experimental results seems to be based on the tenuous assumption that the low-cost particulate matter sensors all function as aerosol spectrometers rather than as nephelometers. In the GRIMM 1.108 aerosol spectrometer, sample air is aerodynamically focused and passes through a narrowly-focused laser beam so that the detector sees the light scattered by just one particle at a time. Thus, the GRIM1.108 can count and size individual particles. The low-cost sensors evaluated in this study do not aerodynamically focus the sample air and, as a result, it is unlikely that the detectors in these sensors see light scattered by individual particles. Instead, the mass concentrations reported by the low-cost sensors are most likely determined by the intensity of light scattered by a group of particles, and not from a measured particle count and size distribution. Results presented by Kelly et al. in 2017 (doi: 10.1016/j.envpol.2016.12.039) suggested that, for Plantower sensors, "the allocation of light scattering to PM1, PM2.5, and PM10 is based on a theoretical model rather than a measurement" (pp. 495-496) and "the size distribution provided by the PMS is based on a theoretical model rather than a measurement" (pp. 497). Claims in the literature that sensors such as the PMS5003 do function as optical particle counters are dubious.

2. Lines 207: I would not say that "the particle sizing of the different bins is not working properly." See the comment above for justification. I think it would be appropriate to say "it is apparent that the PMS5003 does not accurately distinguish between the PM1, PM2.5, and PM10 size fractions." Do the ratios of the masses in the PM1, PM2.5, and PM10 change as the sensor are exposed to different particle sizes?

3. Line 223: Levy Zamora et al. evaluated the PMSA003 sensor, so results from that study might not be directly comparable to results from this study.

4. Lines 231-235: The authors state that they expect the SDS011 to measure PM2.5 more accurately than the PMS5003 based on differences in 2.5 – 10 um mass fractions reported by the two sensors. I don't see how differences in the 2.5 – 10 um mass fractions would affect the reported 0.3 – 2.5 um mass fractions. The two sensors have similar detection efficiencies for the 0.3 – 2.5 um mass fraction. How does the accuracy of PM2.5 measurements reported by the SDS011 (as reported by Badura et al. and Liu et al.) compare to the accuracy of PM2.5 measurements reported by the PMS5003 (see, for example, Malings et al., doi: 10.1080/02786826.2019.1623863)?

5. Lines 240-241: The SPS30 product datasheet states that the SPS30 is calibrated using "a defined potassium chloride particle distribution" (www.sensirion.com/fileadmin/user_upload/customers/sensirion/Dokumente/0_Datasheets/Particulate_Matter/Sensirion_

6. Lines 246-247: The $R^2$ values reported here provide no indication of sensor accuracy. Accuracy is related to the absolute difference between the concentration reported a low-cost sensor and a reference monitor. It's possible for the mass concentrations reported by a low-cost sensor to be very different in magnitude from the concentrations reported by an FEM monitor but for the concentrations reported by the two to still be strongly correlated. Saying that the fraction of variance explained by a linear model is high doesn't imply that the model has an intercept of zero and a slope of one.

7. Lines 281-288: Again, the $R^2$ values reported here provide zero information on sensor accuracy. These values only imply whether or not a linear model explains the relationship between concentrations reported by a reference monitor and concentrations reported by the sensor. Measures of accuracy include mean error, mean absolute error, and mean percentage error.

8. Lines 309-317: See comment #1 on the results.

Comments on organization:

1. Figure 3: This figure could be improved to help readers interpret it. A title above

the legend that says "Particle size bins" would be nice. I also suggest revising the first sentence in the caption to something like "Normalized detection efficiency of the 15 particle size bins as a function of the count median diameter of the reference aerosol." Finally, please also move the sentences from lines 185-186 ("For the sake of clarity. . .") and line 187-188 ("Bins 14 and 15,. . .") to the figure caption.

2. Figure 4: The caption for this figure should be more informative. I suggest "Normalized detection efficiency of discretized PM mass fractions reported by the low-cost sensors as a function of the count median diameter of the reference aerosol." I also suggest moving the text on lines 201-204 ("The coloured circles represent. . ...ranges stated by the corresponding manufacturer.") to the caption.

Editorial comments:

1. Line 13: The PM acronym should come before the word 'sensors'.

2. Lines 17-18: I disagree with the statement "This implies that there are underlying reasons yet to be characterized which are causing inaccuracies in sensor measurements." The underlying reasons for why light-scattering sensors are inaccurate (regardless of cost) has been known among the aerosol research community for decades; there is a vast literature on the limitations and biases associated with this measurement method.

3. Line 29: "enabling" should be "enable" and "higher resolution" should be "higher-resolution".

4. Lines 30-31: It might be helpful to refer readers to some more recent studies demonstrating applications of sensor networks. See, for example, Feinberg et al., 2019 (doi: 10.1016/j.atmosenv.2019.06.026) and Rickenbacker et al., 2019 (doi: 10.1016/j.scs.2019.101473).

5. Line 54: I would use a phrase like "size discrimination" instead of "size selectivity" in this context. The term "size selectivity" is most often used in the aerosol literature

to discuss inlet aspiration/transmission performance, such as that of a PM10 or PM2.5 inertial separator.

6. Line 59: "cursory" should be "cursorily" or "concurrently", depending on what the authors mean by this sentence.

7. Line 82: Rephrase as "However, the more stable sample flow system (i.e., fan instead of convection) might help compensate for the sub-optimal layouts of these sensors."

8. Line 85: "analogue" should be "analog".

9. Line 86: For clarity, please change "sensor outputs shown" to "PM mass fractions listed".

10. Line 92: Change "being" to "to be" and delete the word "other". The GRIMM 1.108 does not measure PM mass directly. Also, please specify which metric reported by the GRIMM 1.108 (or derived from values reported by the GRIMM 1.108) has a comparable accuracy to the filter weighing method.

11. Line 126: Delete the word "produced".

12. Lines 227-228: Consider replacing "two more clear different" with "two clearly different".

13. Line 242: Revise as "and neither Web of Science nor Scopus literature research showed...".

14. Line 249: Replace "align" with "alignment".

15. Line 293: Replace "prominent" with "promising".

16. Line 296: Add the word "Neither" before "Web of Science".

17. Line 307: Replace "pertain" with "retain".

18. Figure 2: The title for this graph is "Sample aerosol", but the aerosol sampled by the

low-cost sensors is referred to as the "reference aerosol" throughout the manuscript. Please use consistent terminology.

References

GRIMM 1.108 manual: https://wmo-gaw-wcc-aerosol-physics.org/files/opc-grimm-model–1.108-and-1.109.pdf

---

## Author Comment (AC1) · 28 Feb 2020

*General Comments:*

*"This manuscript describes the development of a system to provide a flow of monodisperse particles, and the authors then use this system to evaluate several common low-cost sensors. The description and evaluation of the new system (VOAG-GP50) are clearly described. Although their conclusion that low-cost sensors mis-classify particle sizes is not new, the systematic evaluation of size selectivity in low-cost sensors is a valuable contribution to the field. However, the paper would be stronger if the authors improved the clarity of the data-processing section, address a few questions regarding sample flow rate, and polish the language."*

We thank you for your feedback. Several improvements have been made to the manuscript according to the more detailed comments.

*Specific comments:*

*"The data processing section needs some clarification. The authors discuss dividing the data into 30 size bins, but in line 146 they discuss 10 steps to produce different particle sizes. The authors need to clarify how 10 steps can yield 30 different size bins."*

The 10-step program (or method) refers to the dispensing of liquids and is related to neither data processing nor the amount of different particle sizes produced. The number of used steps and the parameters assigned to them simply define the minimum and maximum particle size and the rate at which the particle size gradient evolves from the minimum size to maximum size. The word "gradient" is used to underline that a step from e.g. 2 to 3 µm does not in fact lead to a discontinuous and sudden step from one particle size to another.

The 30 bin sizes (or sections) refer to data processing in which the raw 10-second resolution data (typically ~ 300 data points altogether) was divided into 30 "sections" according to the measured CMDs. For each section, an average detection efficiency and CMD was calculated.

To make this clearer, details and a step-by-step example (with figures) of how the data was processed was added to the supplementary material.

Added to manuscript section 2.3 "Data processing": "A detailed example how the data was processed and how the valid detection ranges were calculated is shown in the supplementary material."

Added to manuscript section 2.2.2 "Sampling configuration": "It is worth underlining that the number of steps used in the GP50 dispensing program does not dictate the number of different particle sizes produced. The number of steps and the parameters assigned to them simply define the minimum (blending ratio of the first step) and maximum (blending ratio of the last step) particle size and the rate (step duration) at which the particle size gradient evolves from the minimum size to maximum size. The word "gradient" is used to note that a step from 2 to 3 µm, for instance, does not lead to a discontinuous and sudden step from one particle size to another."

*"In Lines 175 to 180, the authors base their discussion of valid detection ranges on a detection efficiency curve, but I could not find a discussion of how they define a detection efficiency curve. I would suggest providing an example in the supplementary material and illustrating how the upper half of this curve is defined."*

The normalized detection efficiency curve is defined in Eq. 3 and the respective curves are shown in Figure 4a-f. A demonstration how the valid detection ranges were calculated has been added to the supplementary material.

*"I have some concerns regarding the effect of sample flowrate on the low-cost sensors. In the experimental setup, a pump draws the monodisperse particles into the sensor housing at a flow rate of 1 lpm. Figure S2 shows the sensor housing and placement of the low-cost sensors with the flow directed at the sensor inlet (mostly). The authors should consider whether this setup may be skewing their results. This is particularly important for sensors with fans that are designed to operate at a specific flowrate. It is possible that pushing a flowrate that differs from the design flow rate could alter the results. For example, the PMS sensor has a volumetric flowrate of approximately 0.1lpm (which is 10x lower than the volumetric flowrate into the sensor housing). Granted not all of the 1 lpm would flow into the sensor, but this is worth considering."*

There is no clear theoretical basis as to why a different flow rate would change the way the sensor discriminates different particle sizes. Assuming the sensors function as spectrometers (which the results of this study suggest they do, at least partly), the size discrimination of such devices is predicated on the analysis of pulse height caused by the scattering light of particle (Mie-theory). Change in the flow rate would change the frequency (i.e. number concentration) and duration of pulses but not their height. Whether the measured absolute concentrations were higher than expected is trivial as the data analysis of this study was based on normalized concentrations. Although possible, the effect of particle-size dependent sampling losses was originally estimated to be negligible, and as all the tested sensors seemed to exhibit similar size discrimination characteristics as what previous studies had shown, the effect of ancillary flow rate was not addressed in any way.

However, to ensure that the ancillary flow rate did not affect the results, additional tests were conducted with flow rates of 0.5 and 2 L min$^{-1}$. Instead of testing all the three sensor units of the six different sensor models, only a single unit (unit #3) for each sensor model was evaluated. The results (Fig. 1, attachment) indicate that different flow rates had no meaningful effect on the responses. The SDS011 shows slightly stronger response for particles larger than ~2 – 3 μm, but this is probably resulting from operator inconsistency (or randomness) because the change is similar for both 0.5 and 2 L min$^{-1}$ flow rates. The B5W sensor has weaker response for particle sizes larger than ~4 – 5 μm with 2 L min$^{-1}$ flow rate which suggests that the sampling losses may have increased. However, the response is similar for 0.5 and 1 L min$^{-1}$ flow rates (B5W was originally designed to be used with a heater resistor-induced flow which is most probably closer to 0.5 than 2 L min$^{-1}$). The difference in PPD42NS responses, which imply that the losses may have increased for smaller and not higher flow rates, is attributed to randomness.

While conducting the additional flow rate tests, the Sharp sensors were found to exhibit completely different characteristics to what was previously measured. The underlying reason for this is still unknown, but it is possible that a prototyping breadboard, which was used to make the required connections for the external resistor and capacitor, had loose connections which resulted in misleading bias measurements. Nevertheless, all the Sharp sensors were re-evaluated with the original 1 L min$^{-1}$ flow rate and the unit #3 was tested additionally with the 0.5 and 2 L min$^{-1}$ flow rates. The new valid detection range was measured to be < 0.8 μm which is no longer in an obvious conflict with the results of the previously mentioned study of Wang et al. (2015). Considering the additional tests, it appears that the different flow rates may influence the sensor response in smallest particle sizes (< 0.55 μm), but the responses with 0.5 and 1 L min$^{-1}$ flow rates are so similar that the stated valid detection range remains the same. Smaller flow rates are likely to better represent the original flow rate, which for the Sharp sensors, was based on plain diffusion.

Added to manuscript section 2.2.2 "Sampling configuration": "Although there is no clear theoretical basis as to why a different flow rate would affect the way the sensor discriminates different particle sizes (apart from the different particle size-specific sampling losses), additional tests were conducted with flow rates of 0.5 and 2 L min-1 to ensure that this was indeed the case (see Supplemental Figure S2)."

Section regarding the results of Sharp GP2Y1010AU0F has been revised as: "The response function of the GP2Y1010AU0F is shown in Figure 4d, and its valid detection range appears to be approximately < 0.8 μm. Like the previously discussed sensors, the GP2Y1010AU0F can be used to measure small particles (e.g., PM1) but not coarse mode particles. Several laboratory evaluations have been previously conducted for the GP2Y1010AU0F, but none of these have assessed its detection range using monodisperse test aerosols (Li and Biswas, 2017; Manikonda et al., 2016; Sousan et al., 2016). Wang et al. (2015) used atomized polystyrene latex (PSL) particles to evaluate the effect of particle size on the GP2Y1010AU0F response, but no concluding remarks can be obtained from these results. The study method utilized only three different sized PSLs; moreover, it was not designed to investigate the complete detection range of the GP2Y1010AU0F. However, according to the authors, the results implied that the sensor was more sensitive to 300 nm particles than to 600 and 900 nm particles, which is in slight disagreement with the results of this study whereby the normalized detection efficiency curve shows the highest sensitivity peak for 0.6 μm sized particles as well as a decreasing trend for particles smaller than this. There is no obvious explanation for this discrepancy, but it is worth re-emphasizing the differences in the used evaluation approaches."

[Figure]

Fig 1. Results of the additional tests.

*"The manuscript needs a thorough review and edit by a native English speaker. The language is awkward and sometimes confusing. I am including a few examples from the abstract, but this list is not comprehensive: - "due to their prospective nature regarding spatial extension of measurement coverage". Vague and awkward wording. - "sensors can be useful in achieving this goal". No goal is mentioned previously. - "it is often reminded that the risk of sensor misuse". Improper usage."*

Commercial editing services were used to check and correct the language.

*Technical corrections:*

*"Line 121. Do the authors mean stable particle size distribution rather than particle size gradient? If they mean particle size gradient, this needs to be explained."*

This has been rephrased as: "The novelty of the aerosol generation method used in this research is based on the observation that the particle size of the monodisperse and constant number concentration reference aerosol can be controlled by feeding solutions with different non-volatile concentrations to the VOAG, one after each other" (as suggested by Referee #1).

*"Line 185. The authors should clarify what the response curves are for. The 10-step(30-bin) generation of monodisperse particle sizes?"*

The GRIMM 1.108 was tested the same way as was all the low-cost sensors. The goal was to show how the particle size discrimination characteristics of the mid-cost, 15 bin spectrometer-type instrument differed from the ones of the low-cost sensors.

Following changes were made to the manuscript: "The response curves" replaced with "The normalized detection efficiencies of the 15 bin GRIMM 1.108…"

*"Line 203. The authors should provide the standard deviations of the CMDs. In Figure2, the text says, "The GSD of the size distribution remains below 1.2, but line 113 says that the VOAG has relative standard deviation of less than 3 %.""*

The reason not to provide standard deviations of CMDs is discussed in the added supplementary material. In short, they were insignificant.

The statement "… standard deviation of less than 3 %" refers to the standard deviation of the **number concentration** of the VOAG, which was stated in the original paper of Berglund and Liu (1973), and it is in no way related to **size distributions**.

Line 113 rephrased as: "According to Berglund and Liu (1973), the output aerosol number concentration of the VOAG has a relative standard deviation of less than 3 %, and the formed particle size distribution is monodisperse having a geometric standard deviation (GSD) less than 1.014".

*"The authors mention the size limitation of the APS as being a limiting factor in the analysis (Figure 2), but in line 119 they mention that the VOAG cannot reliably generate particles smaller than 0.55 um. This limitation should be mentioned in Figure 2 in addition to the APS limitation. It would also be worth mentioning this important limitation in the abstract."*

Added to line 19 (Abstract): "(from ~ 0.55 to 8.4 µm)"

Added to Figure 2 caption: "Along with the lower detection limit of the APS, another limiting factor of the study was the smallest producible particle size, which was approximately 0.55 µm."

*"Figure 1 – The GRIMM is not shown. Where does the GRIMM draw its sample? How are the flows distributed symmetrically between the APS and GRIMM since the GRIMM's flow is 1.2 lpm whereas the GRIMM is 1 lpm?"*

The GRIMM drew its sample from where the sensor enclosure is now shown. The isokinetic flow splitter was designed for equal (1 L min$^{-1}$) flow rates, but this was not considered problematic as there was never intention to evaluate the GRIMM other than cursorily. Furthermore, the Figure 3 shows that the unequal flow distribution probably had little to no effect on the response.

Figure 1 caption rephrased as: "Figure 1: Schematic of the sensor evaluation setup. The GRIMM 1.108 drew its sample from where the sensor enclosure is now shown."

---

## Author Comment (AC2) · 28 Feb 2020

*"In this work, Kuula et al. present an investigation of counting efficiency as a function of size for a series of low-cost light scattering devices. The research question that the authors seek to answer (whether low-cost sensors can correctly assign particles to the appropriate PM mass fractions) is an important one. The work appears well thought out and carefully conducted, although the manuscript is below average from an English language perspective. There are dozens of grammatical and stylistic mistakes that will need attention prior to publication. However, the authors present a novel experimental procedure and some interesting results, so I am supportive of publication if this and other concerns are addressed in a revised manuscript. Comments on how the presentation of the methods and the interpretation of the results could be improved are below."*

We thank you for your constructive feedback. Several improvements have been made to the manuscript according to the more detailed comments.

*Major Comments:*

*"One major drawback of this work is that the authors appear to have used a "forced flow" of 1 L/min through each sensor, which may not be in accordance with the manufacturer's intended use. The use of a non-standard flow rate (i.e., one that is different from the manufacturer's recommendation) could compromise a sensor in a number of ways. For example, it could lead to differential inertial losses of particles as they transmit through the sensor. Or, it could lead to variable pulse widths of particles during detection that deviate from the expected pulse widths during calibration. The authors need to verify that this method of testing under a forced flowrate did not affect sensor response. One way would be to repeat these tests at flow rates of 0.5 and 2 Lpm to see if the response factors change. If they do change, then the results presented here might lead to misinterpretation...I do not support publication of this manuscript until this issue is investigated further and/or resolved."*

Below is the copy of our response to Referee #2 (concerning the same question):

There is no clear theoretical basis as to why a different flow rate would change the way the sensor discriminates different particle sizes. Assuming the sensors function as spectrometers (which the results of this study suggest they do, at least partly), the size discrimination of such devices is predicated on the analysis of pulse height caused by the scattering light of particle (Mie-theory). Change in the flow rate would change the frequency (i.e. number concentration) and duration of pulses but not their height. Whether the measured absolute concentrations were higher than expected is trivial as the data analysis of this study was based on normalized concentrations. Although possible, the effect of particle-size dependent sampling losses was originally estimated to be negligible, and as all the tested sensors seemed to exhibit similar size discrimination characteristics as what previous studies had shown, the effect of ancillary flow rate was not addressed in any way.

However, to ensure that the ancillary flow rate did not affect the results, additional tests were conducted with flow rates of 0.5 and 2 L min$^{-1}$. Instead of testing all the three sensor units of the six different sensor models, only a single unit (unit #3) for each sensor model was evaluated. The results (Fig. 1, attachment) indicate that different flow rates had no meaningful effect on the responses. The SDS011 shows slightly stronger response for particles larger than ~2 – 3 μm, but this is probably resulting from operator inconsistency (or randomness) because the change is similar for both 0.5 and 2 L min$^{-1}$ flow rates. The B5W sensor has weaker response for particle sizes larger than ~4 – 5 μm with 2 L min$^{-1}$ flow rate which suggests that the sampling losses may have increased. However, the response

is similar for 0.5 and 1 L min$^{-1}$ flow rates (B5W was originally designed to be used with a heater resistor-induced flow which is most probably closer to 0.5 than 2 L min$^{-1}$). The difference in PPD42NS responses, which imply that the losses may have increased for smaller and not higher flow rates, is attributed to randomness.

While conducting the additional flow rate tests, the Sharp sensors were found to exhibit completely different characteristics to what was previously measured. The underlying reason for this is still unknown, but it is possible that a prototyping breadboard, which was used to make the required connections for the external resistor and capacitor, had loose connections which resulted in misleading bias measurements. Nevertheless, all the Sharp sensors were re-evaluated with the original 1 L min$^{-1}$ flow rate and the unit #3 was tested additionally with the 0.5 and 2 L min$^{-1}$ flow rates. The new valid detection range was measured to be < 0.8 µm which is no longer in an obvious conflict with the results of the previously mentioned study of Wang et al. (2015). Considering the additional tests, it appears that the different flow rates may influence the sensor response in smallest particle sizes (< 0.55 µm), but the responses with 0.5 and 1 L min$^{-1}$ flow rates are so similar that the stated valid detection range remains the same. Smaller flow rates are likely to better represent the original flow rate, which for the Sharp sensors, was based on plain diffusion.

Added to manuscript section 2.2.2 "Sampling configuration": "Although there is no clear theoretical basis as to why a different flow rate would affect the way the sensor discriminates different particle sizes (apart from the different particle size-specific sampling losses), additional tests were conducted with flow rates of 0.5 and 2 L min-1 to ensure that this was indeed the case (see Supplemental Figure S2)."

Section regarding the results of Sharp GP2Y1010AU0F has been revised as: "The response function of the GP2Y1010AU0F is shown in Figure 4d, and its valid detection range appears to be approximately < 0.8 µm. Like the previously discussed sensors, the GP2Y1010AU0F can be used to measure small particles (e.g., PM1) but not coarse mode particles. Several laboratory evaluations have been previously conducted for the GP2Y1010AU0F, but none of these have assessed its detection range using monodisperse test aerosols (Li and Biswas, 2017; Manikonda et al., 2016; Sousan et al., 2016). Wang et al. (2015) used atomized polystyrene latex (PSL) particles to evaluate the effect of particle size on the GP2Y1010AU0F response, but no concluding remarks can be obtained from these results. The study method utilized only three different sized PSLs; moreover, it was not designed to investigate the complete detection range of the GP2Y1010AU0F. However, according to the authors, the results implied that the sensor was more sensitive to 300 nm particles than to 600 and 900 nm particles, which is in slight disagreement with the results of this study whereby the normalized detection efficiency curve shows the highest sensitivity peak for 0.6 µm sized particles as well as a decreasing trend for particles smaller than this. There is no obvious explanation for this discrepancy, but it is worth re-emphasizing the differences in the used evaluation approaches."

[Figure]

Fig 1. Results of the additional tests.

*"I think the VOAG method for generating a sequence of monodisperse particles of varying size is both novel and useful to the community. However, I would appreciate more detail on how the method is implemented so that others can properly reproduce this method. Such details could be enumerated as supplemental material."*

All necessary running parameters of the VOAG and GP50 have been presented in the supplementary material Tables S1-2.

*Comments on the Methods section:*

*"The Methods section could be organized better and was missing key information needed to reproduce the work."*

See our previous comment. Improvements to the Methods section have been made according to the referee comments listed in the next sections.

*Comments on technical issues:*

*"1. Table 1: The "number of size bins" column is problematic. It's not appropriate to compare the number of PM mass fractions reported by the low-cost sensors to the number of size bins that the GRIMM 1.108 classifies particle into. Perhaps the authors could remedy this problem by creating two columns: one labeled "number of mass fractions reported" and "number of particle size bins". For example, the SPS30 reports data in 4 mass fractions and 5 particle size bins."*

Due to space limitations, the column "Number of size bins" was rephrased as "Number of mass fractions reported" but no additional column was added. Column "Sensor output" was modified for the GRIMM as "All fractions individually".

*"2. Line 76: Why did the authors remove the mechanism that drives sample flow through some of the sensors but not others? Specifically, why did they remove the air heating resistors from the PPD42 and B5W but not remove the fans from the PMS5003, SDS011, and SPS30? This discrepancy seems like it could lead to an unfair comparison between different sensors. A sensor's output is partially determined by how quickly and efficiently particles are drawn into the sensing zone. If the flow rate through the sensor is modified, the output of the modified sensor might differ from the output of the "as-purchased" sensor. For example, results presented by Tryner et al. (doi:10.1039/C9EM00234K) indicate that varying the flow rate through a modified PMS5003sensor changes the output. Since the authors modified some of the sensors used in this study, the results presented here might not be directly transferrable to unmodified sensors being used in the field."*

All decisions regarding the used sampling setups were based on two factors. Most importantly, the configuration had to be such that the reference aerosol remained representative and valid throughout the test run. A secondary factor was that the configuration had to resemble the original configuration as closely as possible.

When relied on convection, the sensor must be orientated in an upright position, and the sample needs to be fed to the sensor from the bottom. This poses practical issues which are hard to overcome. For example, it is unclear how a fresh and representative sample could be provided to the proximity of the resistor without disturbing the temperature gradient and/or the flow rate of the convection itself. The feeding flow rate would have to be very small in order to avoid any disturbance, but this would simultaneously compromise the explicitness of the reference aerosol. The use of primary standard reference particles was one of the most important elements of this study. Regarding the PPD42NS specifically, if the resistor-demanded orientation is used, the lenses of this sensor are facing towards the incoming flow (forward angle scattering) which, consequently, exposes them to fouling (drift in sensitivity). This is a major problem, which we encountered in our previous paper (10.3390/s17122915), and there is no clear solution for this apart from reversing the flow direction. On a general note, the convection-based flow is poorly suited for any kind of sampling due to its susceptibility to external disturbances, such as the dynamic effects of wind and changes in ambient temperature.

Removing the fans from the PMS5003 and SPS30 would most probably lead to significant sampling losses due to their 90-degree elbows (and no mechanical flow through the sensor), and the obvious way to account for this would be to relocate the intake (drill an alternative hole to the sensor body) so that the sample is introduced directly to the detection zone. However, this is most certainly not the way the sensors are typically used. The SDS011 would probably function acceptably even without its fan, but it was considered better to retain the original configuration.

We agree that the testing setups are not perfect and that currently, some of the sensors are probably used in a manner that they were not designed for. However, creating a uniform experiment configuration for sensors which all entail different type (and poorly designed) layouts and flow rates will always be a compromise of some sort. To the extent that the custom sampling setups may have affected the outcome, we want to underline the main finding of this study is that the sensors do not measure particle sizes which their technical specifications imply, and that the ancillary flow rates, which were discussed earlier, did not have a significant impact on the results.

Study of Tryner et al. is not helpful as the particle size discrimination of PMS5003 was not investigated; it likely that the concentrations increased across all outputs respectively.

*"3. Lines 85-89: The PMS5003 reports each mass fraction two ways: "CF=1, standard particle" and "under atmospheric environment". Which values were used in the data analyses presented here?"*

Added to Table 1 notes: "Standard particle (CF=1) output was used."

*"4. Lines 164-165: Please state the fraction of raw measurement points that were disregarded."*

Added: "(~ 2.1 % of the data)"

*"5. Line 170: Equation 3 is not clear. What is the maximum Sensor/APS ratio? Was this the maximum ratio measured during the entire 60-minute long test run?"*

Maximum Sensor/APS ratio refers to the maximum Sensor/APS ratio measured during a single test run.

Added: "Maximum Sensor/APS ratio refers to the maximum ratio measured during a single test run."

*"6. Line 175: One issue is that the APS is known to have poor counting efficiency for liquid droplets above about 5 microns in diameter. Did the authors account for this artifact? See [https://doi.org/10.1016/j.jaerosci.2005.03.009](https://doi.org/10.1016/j.jaerosci.2005.03.009)"*

This artifact was not accounted.

Added to section 2.2.2 "Sampling configuration": "Although the reference instrument APS is known for having decreased counting efficiency for liquid droplets over ~5 µm in size (Volckens and Peters, 2005), no additional corrections were used."

*"7. Line 179-180: This definition of "valid detection ranges" is not clear. Do the authors mean that the valid detection range was the range of CMDs for which the normalized detection efficiency was greater than 50%? If so, please consider also rephrasing the text on lines 209-210 as something like "(the valid detection range was defined as the range of particle sizes for which normalized detection efficiency was > 50%)"."*

Definition of > 50 % was not used because the obtained detection efficiency ranges were different for different sensor models (e.g. not from 0 to 100 %) and the normalized detection efficiency does not

describe the proportion of measured concentrations. A detailed example how the upper half of the normalized detection efficiency curve was calculated has been added to the supplementary material.

Added to section 2.3 "Data processing": "A detailed example how the data was processed and how the valid detection ranges were calculated is shown in the Supplementary Material".

*Comments of organization:*

*"1. Lines 75-84: This information should be moved to the "Sampling configuration" section. These methods are difficult to understand or justify without additional knowledge of the system used to pass monodisperse aerosol through the sensor."*

This paragraph was moved to section 2.2.2 "Sampling configuration".

*"2. Lines 85-82: This information should be moved to the "Data processing" section."*

It is assumed that the referee means lines 85-89 (otherwise it contradicts the comment above). This information was moved to section 2.3 "Data processing".

*"3. Lines 89-90: Please move the sentence that starts "Three units of each sensor..."to the end of the first paragraph in Section 2.1."*

Corrected.

*"4. Lines 90-93: Please start a new paragraph here that contains the information on the GRIMM 1.108."*

Corrected.

*"5. Line 95: Section 2.2.1 should start with a sentence that says, "The aerosol sampled by the low-cost sensors was generated using a Vibrating Orifice Aerosol Generator3450 (VOAG, TSI Inc., USA)." That way readers will know why the information contained in this section is relevant."*

Corrected.

*"6. Lines 101-102, "This aerosol generation method...": This sentence is neither relevant nor helpful. Please delete it."*

Deleted.

*"7. Lines 121-125: The second sentence in this paragraph is more helpful than the first. I suggest rephrasing as "The novelty of the aerosol generation method used in this research is based on the observation that the particle size of the monodisperse and constant number concentration reference aerosol can be controlled by feeding solutions with different non-volatile concentrations to the VOAG, one after each other.""*

Rephrased as suggested.

*"8. Lines 161-162: Please show at least one example panel from Figure S2 in the main text."*

PMS5003 panel added to manuscript: "An illustration of the PMS5003 sampling arrangement is shown in Figure 3."

*"9. Lines 176-177, "The decision to divide the data into 30 bins...": This sentence is not necessary. Please consider deleting it."*

Deleted.

*"10. Lines 186-187: The sentence that begins "The previously-described data processing technique..." should be moved to the "Data processing" section."*

Moved.

*Comments of the Results section*

*Comments on technical issues:*

*"1. Overall, the authors interpretation of their experimental results seems to be based on the tenuous assumption that the low-cost particulate matter sensors all function as aerosol spectrometers rather than as nephelometers. In the GRIMM 1.108 aerosol spectrometer, sample air is aerodynamically focused and passes through a narrowly focused laser beam so that the detector sees the light scattered by just one particle at a time. Thus, the GRIM1.108 can count and size individual particles. The low-cost sensors evaluated in this study do not aerodynamically focus the sample air and, as a result, it is unlikely that the detectors in these sensors see light scattered by individual particles. Instead, the mass concentrations reported by the low-cost sensors are most likely determined by the intensity of light scattered by a group of particles, and not from a measured particle count and size distribution."*

Whether the sensors function as spectrometers, nephelometers, or as a combination of these two is unknown, and a thorough reverse engineering of the electronic design (and software) would be required in order to reach ultimate conclusion. According to the results of this study, the sensors exhibit some kind of size discrimination capability and this observation in an agreement with the broader scientific community which uses sensors to measure size-specific mass fractions, such as the PM2.5. Whether the sensors function solely as spectrometers, or whether they are switching between single particle detection and total scattered light intensity measurement, remains unknown. It is worth noting that nephelometers, by design, do not have a mechanism (apart from size-selective inlets) which would allow them to size discriminate individual particles, and thus, the fact that many of the sensors evaluated in this study showed little to no response in larger particle sizes would be unexplainable. According to Mie-theory, the total intensity of the scattered light is proportional to the sixth power of particle diameter.

To avoid particle coincidence, the optical detection layout does not need to be complicated. For example, a detection volume of 1 cm$^3$ (e.g. 10 x 10 x 10 mm cube) would imply that, statistically, the concentration limit for particle coincidence is 1 cm$^{-3}$. With detection volume of 0.125 cm$^3$ (e.g. 5 x 5 x 5 mm cube) the respective concentration limit is 8 cm$^{-3}$. Typical number concentrations for coarse mode (1–10 μm) particles are relatively low, for instance, 7 ± 9 cm$^{-3}$ in urban Beijing (Wu et al. 2008 10.1016/j.atmosenv.2008.06.022) and 2 ± 3 cm$^{-3}$ in regional China (Shen et al. 2011, 10.5194/acp-11-1565-2011), and therefore, it is reasonable to assume that a fairly simple layout capable of measuring individual particles (at least occasionally) without additional optical or aerodynamic lenses is possible to engineer and design.

*"Results presented by Kelly et al. in 2017 (doi: 10.1016/j.envpol.2016.12.039) suggested that, for Plantower sensors, "the allocation of light scattering to PM1, PM2.5, and PM10 is based on a theoretical model rather than a measurement" (pp. 495-496) and "the size distribution provided by the PMS is based on a theoretical model rather than a measurement" (pp. 497). Claims in the literature that sensors such as the PMS5003 do function as optical particle counters are dubious."*

The conclusion by Kelly et al. is most probably right and it is in an agreement with the results of this study. But it does not provide evidence that the PMS works solely as a nephelometer.

*"2. Lines 207: I would not say that "the particle sizing of the different bins is not working properly."
See the comment above for justification.  I think it would be appropriate to say, "it is apparent that the
PMS5003 does not accurately distinguish between the PM1, PM2.5, and PM10 size fractions." Do the
ratios of the masses in the PM1, PM2.5, and PM10 change as the sensor are exposed to different
particle sizes?"*

Line 207 corrected as suggested. Whether the ratio of mass changes or stays constant does not
determine the difference or similarity in size discrimination capability. For example, two outputs with
identical size discrimination could have different mass ratios if their relation is exponential (e.g. PM1=
PM2.5^2). For the PMS, the ratios were mostly the same; PM1/PM2.5 (3.7±0.4), PM2.5/PM10
(1.6±0.4), PM1/PM10 (5.8±0.9).

*"3.  Line 223: Levy Zamora et al. evaluated the PMSA003 sensor, so results from that study might not
be directly comparable to results from this study."*

This reference was removed.

*"4.  Lines 231-235:  The authors state that they expect the SDS011 to measure PM2.5 more accurately
than the PMS5003 based on differences in 2.5 – 10 um mass fractions reported by the two sensors.  I
don't see how differences in the 2.5 – 10 um mass fractions would affect the reported 0.3 – 2.5 um
mass fractions. The two sensors have similar detection efficiencies for the 0.3 – 2.5 um mass fraction.
How does the accuracy of PM2.5 measurements reported by the SDS011 (as reported by Badura et al.
and Liu et al.) compare to the accuracy of PM2.5 measurements reported by the PMS5003(see, for
example, Malings et al., doi: 10.1080/02786826.2019.1623863)?"*

The statement is based on the notion that in practice, the SDS011 has two different outputs for ~ 0.3
– 2.5 µm range (approximately < 0.8 and 0.7 – 1.7 µm) whereas the PMS has only one (all three outputs
seem to measure similar size fractions; < 0.7, < 0.8, and < 1 µm). By utilizing both outputs of the SDS011
(for example, by calculating the ratio of the bins), a more accurate approximation for the mass
distribution in 0.3 – 2.5 µm range, and thus higher accuracy, is possible to obtain. Of course, the user
of the sensor must be aware of the particular size discrimination characteristics of the SDS011 (e.g.
PM10 does not measure PM10).

To make this clearer, the statement has been rephrased as: "For example, by calculating the ratio of
bins 1 and 2, it is possible to approximate the distribution of mass in the 0.3–2.5 µm size range, thus
using an additional correction factor to obtain more accurate results."

*"5. Lines 240-241: The SPS30 product datasheet states that the SPS30 is calibrated using "a defined
potassium chloride particle distribution"."*

([www.sensirion.com/fileadmin/user_upload/customers/sensirion/Dokumente/0_Datasheets/Partic
ulate_Matter/Sensirion_PM_Sensors_SPS30_Datasheet.pdf](www.sensirion.com/fileadmin/user_upload/customers/sensirion/Dokumente/0_Datasheets/Particulate_Matter/Sensirion_PM_Sensors_SPS30_Datasheet.pdf)).

To be precise, the documentation states: "PM2.5 accuracy is verified for every sensor after calibration
using a defined potassium chloride particle distribution". It does not disclose how PM1, PM4 or PM10
have been calibrated. We asked Sensirion about this and the unofficial answer was that PM1 is
calibrated with KCl and PM2.5, PM4 and PM10 with Arizona dust. PM4 and PM10 are just
extrapolations of the PM2.5 response (which is evident according to the results of this work).

*"6. Lines 246-247: The R^2 values reported here provide no indication of sensor accuracy. Accuracy is
related to the absolute difference between the concentration reported a low-cost sensor and a
reference monitor.  It's possible for the mass concentrations reported by a low-cost sensor to be very*

*different in magnitude from the concentrations reported by an FEM monitor but for the concentrations reported by the two to still be strongly correlated. Saying that the fraction of variance explained by a linear model is high doesn't imply that the model has an intercept of zero and a slope of one."*

Word "accuracies" replaced with "coefficient of determinations".

*"7. Lines 281-288: Again, the $R^2$ values reported here provide zero information on sensor accuracy. These values only imply whether a linear model explains the relationship between concentrations reported by a reference monitor and concentrations reported by the sensor. Measures of accuracy include mean error, mean absolute error, and mean percentage error."*

Word "accuracy" replaced with "performance".

*"8. Lines 309-317: See comment #1 on the results."*

See the respective response.

*Comments on organization:*

*"1. Figure 3: This figure could be improved to help readers interpret it. A title above the legend that says "Particle size bins" would be nice. I also suggest revising the first sentence in the caption to something like "Normalized detection efficiency of the 15particle size bins as a function of the count median diameter of the reference aerosol." Finally, please also move the sentences from lines 185-186 ("For the sake of clarity...") and line 187-188 ("Bins 14 and 15...") to the figure caption."*

Legend title added. Caption rephrased as suggested: "Figure 4. Normalized detection efficiency of the 15 particle size bins as a function of the count median diameter of the reference aerosol. Consecutively increasing and decreasing response curves indicate that the particle sizing of the instrument is functioning correctly. For the sake of clarity, degrees of measurement variation have been excluded from the figure. Bins 14 and 15, which correspond to 10–15 and 15–20 µm, respectively, are not shown as they did not produce any response (as expected)."

*"2. Figure 4: The caption for this figure should be more informative. I suggest "Normalized detection efficiency of discretized PM mass fractions reported by the low-cost sensors as a function of the count median diameter of the reference aerosol." I also suggest moving the text on lines 201-204 ("The colored circles represent.... ranges stated by the corresponding manufacturer.") to the caption."*

Caption rephrased as suggested: "Figure 5. Normalized detection efficiency of discretized PM mass fractions reported by the low-cost sensors as a function of the count median diameter of the reference aerosol. The colored circles represent the calculated average responses of the three sensor units, and the shaded background areas represent the respective standard deviations. Standard deviations of the average CMDs were negligible due to the reliable and reproducible test method. Figure legends correspond to the bin size ranges stated by the corresponding manufacturer."

*Editorial comments:*

*"1. Line 13: The PM acronym should come before the word 'sensors'."*

Corrected.

*"2. Lines 17-18: I disagree with the statement "This implies that there are underlying reasons yet to be characterized which are causing inaccuracies in sensor measurements." The underlying reasons for why light-scattering sensors are inaccurate (regardless of cost) has been known among the aerosol*

*research community for decades; there is a vast literature on the limitations and biases associated with this measurement method."*

This statement refers to sensor type measurements specifically, and the results of this study imply that multiple studies have been conducted without understanding what the sensors are truly measuring. We do not consider it appropriate to take a hostile stance in reasoning our study objectives and background.

*"3. Line 29: "enabling" should be "enable" and "higher resolution" should be "higher-resolution"."*

Corrected.

*"4. Lines 30-31: It might be helpful to refer readers to some more recent studies demonstrating applications of sensor networks. See, for example, Feinberg et al., 2019 (doi: 10.1016/j.atmosenv.2019.06.026) and Rickenbacker et al., 2019 (doi:10.1016/j.scs.2019.101473)."*

Study of Feinberg et al. added. Study of Rickenberg et al. was not accessible with our institute credentials and was excluded.

*"5. Line 54: I would use a phrase like "size discrimination" instead of "size selectivity" in this context. The term "size selectivity" is most often used in the aerosol literature to discuss inlet aspiration/transmission performance, such as that of a PM10 or PM2.5inertial separator."*

Corrected.

*"6. Line 59: "cursory" should be "cursorily" or "concurrently", depending on what the authors mean by this sentence."*

Corrected as "cursorily".

*"7. Line 82: Rephrase as "However, the more stable sample flow system (i.e., fan instead of convection) might help compensate for the sub-optimal layouts of these sensors.""*

Corrected.

*"8. Line 85: "analogue" should be "analog"."*

Corrected.

*"9. Line 86: For clarity, please change "sensor outputs shown" to "PM mass fractions listed"."*

Corrected.

*"10. Line 92: Change "being" to "to be" and delete the word "other". The GRIMM1.108 does not measure PM mass directly. Also, please specify which metric reported by the GRIMM 1.108 (or derived from values reported by the GRIMM 1.108) has a comparable accuracy to the filter weighing method."*

Corrected and added: "(mass of C-factor adjusted total suspended particles)"

*"11. Line 126: Delete the word "produced"."*

Corrected.

*"12. Lines 227-228: Consider replacing "two more clear different" with "two clearly different"."*

Corrected.

*"13. Line 242: Revise as "and neither Web of Science nor Scopus literature research showed…"."*

Corrected.

*"14. Line 249: Replace "align" with "alignment"."*

Corrected.

*"15. Line 293: Replace "prominent" with "promising"."*

Corrected.

*"16. Line 296: Add the word "Neither" before "Web of Science"."*

Corrected.

*"17. Line 307: Replace "pertain" with "retain"."*

Corrected.

*"18. Figure 2: The title for this graph is "Sample aerosol", but the aerosol sampled by the ow-cost sensors is referred to as the "reference aerosol" throughout the manuscript. Please use consistent terminology."*

Corrected.

*References*

*"GRIMM 1.108 manual: [https://wmo-gaw-wcc-aerosol-physics.org/files/opc-grimm-model–1.108-and-1.109.pdf](https://wmo-gaw-wcc-aerosol-physics.org/files/opc-grimm-model–1.108-and-1.109.pdf)"*

Added.

---

## Author Response (AR2)

Dear Editor,

we appreciate your feedback, and hopefully the manuscript is now in a clearer form. We are looking forward for your opinion. Below you will find our point-by-point response. The revised manuscript and supplementary material (with 'track changes' on) have also been attached to the electronic submission.

On behalf of all authors,

Joel Kuula
Atmospheric Composition Research
Finnish Meteorological Institute
joel.kuula@fmi.fi

*In the comments below the line numbers refer to the "authors comments" version of the manuscript.*

*I don't think that the response to the reviewers is adequate, in regard to the question of how 10 steps in the program of the GP50 might lead to 30 size bins. To be sure, you have said that the 10 steps are not related to the fact that there are 30 size bins (lines 164 to 169).*

**This is correct; the number of used bins (30) is a computational detail and it is not related to the generation of particles.**

*However the source of this confusion remains. At line 141 you state that the aerosols are monodisperse, and the implication of this sentence is that this is controlled by the GP50.*

**Line 141 implies that the GP50 allows the user to freely choose the produced particle size. To be clear, the fact the aerosol is monodisperse is a feature of the VOAG and not GP50.**

*Thus in the light of your response, one wonders how the different size bins are chosen and/or created. Next at line 158 there is a sentence that still says that the GP50 program (of 10 steps) produced particles sizes in a logarithmically distributed range (0.45 to 9.78 microns). This is still the main clue to the reader as to how you arrived at 30 size bins.*

**The decision to use 30 bins was purely based on the clarity of the produced figures and statistically sufficient number of measurement points (minimum of 3) belonging to each bin. A statement regarding this was included in the original manuscript, but reviewer #1 considered it to be irrelevant. The statement has been added again to the manuscript.**

**Section 2.3 data processing: "The decision to divide the data into 30 bins was based on the clarity of the produced figure and statistically sufficient number of measurement points belonging to each bin."**

*Further, though this may not have been intentional, it could contradict the earlier assertion that your aerosols are monodisperse, since your use of "logarithmically distributed" is ambiguous - it could refer to a continuous statistical distribution. It would be better to say "logarithmically spaced", which is what I think you mean.*

**Term "distributed" replaced with "spaced".**

*Thus I cannot work out where your 30 size bins come from (actually I count 28 bins in figure 5), until much later in the paper at line 204. There I see that the bins are actually determined by the APS measurements. But if you are to get logarithmic spacing something else must predetermine what the mean diameter (CMD) should be. So I'm still puzzled.*

**The reason, why only 28 data points are present in the figures, has been given in the supplementary material; it is because the first and last bin (0.45-0.50 and 8.80-9.73 µm, respectively) of the 30 bins did not in practice contain any measurement points (the size range of the produced particles was approximately 0.55-8.4 µm). By reducing the number of used bins (i.e. widening the width of the bins) it could have been possible to "force" 3 or more data points to each bin; however, this would have compromised the clarity and representativeness of the figures due to the increasing standard deviations and lower size resolution. The lower and upper end of the size range of the 30 bins (0.45 and 9.73 µm, respectively) were chosen according to the theoretical size range of the produced particles. This ensured that the data from different test runs was treated equally.**

**The relevant section of the supplemental material has been appended at the end of this document.**

*Another response that I consider inadequate is in regard to the comment where reviewer 1 raises the possibility of there being detections of multiple particles. It is a off-topic to give expected concentrations in China - what really matters here are the concentrations in your experimental setup, at the detectors. I cannot find any estimate of this number in your revised manuscript. Also the assertion that Mie theory gives the total intensity of the scattered light scaling as the sixth power of diameter is too simple here. The scaling is much more complicated because of the rapidly growing forward scattering lobe (with increasing diameter), so the scaling at a particular angle needs to be considered.*

**The reviewer comment was interpreted to mean that it is unreasonable to assume that the sensors could function as optical particle counters rather than as nephelometers, and that the main reason for this is the particle coincidence resulting from the unsophisticated particle detection configuration. Our response intended to highlight that the stance taken in the manuscript is not, in fact, unreasonable from the technical point of view, and that the assumption of a nephelometer-type functioning is problematic in several different ways; it not only contradicts the previously presented major comment regarding the use of different flow rates, but also undermines all the previous sensor studies, which have attempted to measure size-specific mass fractions. Nephelometers cannot be used to measure sizes of individual particles, and according to the Mie theory, response of a nephelometer type device should be stronger for larger particles and not weaker. This is not what the results of this study showed. The statement regarding the sixth power of diameter is commonly made in aerosol science (see e.g. *W. C. Hinds: Aerosol Technology: Properties, Behavior, and Measurement of Airborne Particles*) and, although being an approximation, it is, in our opinion, sufficient to prove the point in this case.**

**The total number concentration of the reference aerosol is irrelevant considering that the limit value for particle coincidence is not known. The maximum concentration can be estimated from the running parameters of the VOAG (listed in Supplemental Table S1), and, in practice, the concentrations were in the range of 30-90 # cm$^{-3}$ range depending on the particle size and the degree of deposition losses. Compared to other aerosol generators (e.g. atomizers and powder generators), the range of concentrations the VOAG produces is very low.**

**Added to Figure 2 caption: "(concentration range 30-90 # cm$^{-3}$)."**

*Additional comments:*

*Referring to an "optical aerosol spectrometer" could confuse. The spectrum determined is a size spectrum, not an optical spectrum. So saying "optical aerosol size spectrometer" would get around this.*

**Corrected as suggested.**

*At line 154 "The GP50 was operated in a method-mode, meaning that an automated program was used to dispense the liquids." is a bit confusing because "method-mode" is meaningless to me. Better to say "The GP50 used an automated program for dispensing the liquids."*

**Corrected as suggested.**

*In Figure 3, there is nothing that is white, only grey, in spite of the caption referring to something white.*

**Rephrased as "light grey".**

*The explanation of figure 4 needs improvement - in particular, in panel c) there is a legend that presumably refers to size bins as determined by the device under test. Things like this need to be be made more explicit.*

**It is explicitly stated in the figure caption that the figure legends refer to the detection ranges declared by the corresponding manufacturer.**

*At line 286 - South Coast of which country?*

**Added: "USA"**

**Appendix:**

**Supplemental material**

**Detailed description of the data processing method used**

Supplemental Figure S4 shows the normalized and filtered (data points with GSD greater than 1.2 removed) 10-second resolution data of the Omron B5W unit #1 test. Raw data is plotted as transparent bullets and the average values and respective standard deviations (for both CMD and normalized detection efficiency) as solid dots. The raw data was divided into 30 different sections which were logarithmically distributed to 0.45 – 9.73 µm range. This range was the theoretical size range of the produced particles. In the figure, each section corresponds to each solid dot (blue and red), and in this case, a total of 28 dots (for each color) are visible. This is because in practice the first and last section (0.45 – 0.50 and 8.80 – 9.73 µm) did not contain any measurement points.

Despite shown here, the standard deviations of the raw data were not utilized in any form as the final statistical uncertainties were calculated from the average responses of the three individual units. By using the "average of averages", all units had an equal contribution to the final statistics (28 data points each) as in some occasions, the total number of raw data points and the way the points were distributed along different particle sizes varied. See for example the red circle in Fig. S3; for an unknown reason, the speed at which the particle size gradient was evolving decreased momentarily and thus resulted in a cluster of data points. If the raw data would have been used as such, the cluster would have distorted the calculations of average due to the greater number of data points at this specific particle size.

[Figure]

**Supplemental Figure S4. Normalized and filtered (GSDs greater than 1.2 removed) data of the Omron B5W unit #1 test run. The raw 10-sec resolution data is shown as transparent bullets and the calculated average values of the 30 different size sections as solid dots (with standard deviations).**

The average responses of the three Omron B5W units are shown in Supplemental Figure S5. The circle, triangle, and diamond markers stand for the average responses of the individual units #1, #2, and #3,

respectively, and "the average of the averages" (and respective standard deviations) are shown in the figure as star markers. The standard deviations of the average CMDs are negligible compared to the differences observed in normalized detection efficiencies and thus they were not shown in the final manuscript Figure 4f. Supplemental Figure S6, which is essentially the same figure as the final manuscript Figure 4f but with standard deviations of the CMDs included, shows again the insignificance of the CMD standard deviations.

[Figure]

**Supplemental Figure S5. Averaged responses of the three individual sensor units.**

[Figure]

**Supplemental Figure S6. Final normalized detection efficiency of the Omron B5W (with standard deviations).**

---

## Author Response (AR3)

Dear Editor,

the minor typo regarding the figure numbers has been corrected. We wish to thank you as well as the reviewers for taking your time to comment our manuscript. Your feedback was greatly appreciated!

On behalf of all authors,

Joel Kuula
Atmospheric Composition Research
Finnish Meteorological Institute
joel.kuula@fmi.fi